# Levels of Interleukin-6 in Saliva, but Not Plasma, Correlate with Clinical Metrics in Huntington’s Disease Patients and Healthy Control Subjects

**DOI:** 10.3390/ijms21176363

**Published:** 2020-09-02

**Authors:** Jody Corey-Bloom, Ryan S. Fischer, Aeri Kim, Chase Snell, Georgia M. Parkin, Douglas A. Granger, Steven W. Granger, Elizabeth A. Thomas

**Affiliations:** 1Department of Neurosciences, University of California, San Diego, CA 92037, USA; jcoreybloom@health.ucsd.edu (J.C.-B.); cmsnell@health.ucsd.edu (C.S.); 2Salimetrics, LLC, Carlsbad, CA 92008, USA; rfischer@salimetrics.com (R.S.F.); sgranger@salimetrics.com (S.W.G.); 3Institute for Interdisciplinary Salivary Bioscience Research, University of California Irvine, Irvine, CA 92697, USA; gparkin@hs.uci.edu (G.M.P.); dagrange@uci.edu (D.A.G.); 4Department of Epidemiology, University of California Irvine, Irvine, CA 92697, USA; 5Bloomberg School of Public Health, and School of Nursing, Johns Hopkins University, Baltimore, MD 21205, USA

**Keywords:** peripheral, neurodegenerative, cytokine, biomarker, saliva, biofluid

## Abstract

Growing evidence suggests that inflammatory responses, in both the brain and peripheral tissues, contribute to disease pathology in Huntington’s disease (HD), an inherited, progressive neurodegenerative disorder typically affecting adults in their 30–40 s. Hence, studies of inflammation-related markers in peripheral fluids might be useful to better characterize disease features. In this study, we measured levels of C-reactive protein (CRP), Interleukin-6 (IL-6), interleukin 1 beta (IL-1B), and alpha-amylase (AA) in saliva and plasma from *n* = 125 subjects, including *n* = 37 manifest HD patients, *n* = 36 premanifest patients, and *n* = 52 healthy controls, using immunoassays. We found increases in salivary levels of IL-6, IL-1B and CRP across different disease groups and increased levels of IL-6 in the plasma of HD patients as compared to premanifest patients and controls. The levels of salivary IL-6 were significantly correlated with each of the other salivary markers, as well as with IL-6 levels measured in plasma. Further, salivary IL-6 and IL-1B levels were significantly positively correlated with Total Motor Score (TMS) and chorea scores and negatively correlated with Total Functional Capacity (TFC) in HD patients, whereby in healthy control subjects, IL-6 was significantly negatively correlated with Montreal Cognitive Assessment (MoCA) and the Symbol Digit Modalities test (SDM). Interestingly, the plasma levels of IL-6 did not show similar correlations to any clinical measures in either HD or control subjects. These findings suggest that salivary IL-6 is particularly relevant as a potential non-invasive biomarker for HD symptoms. The advent of an effective, dependable salivary biomarker would meet the urgent need for a less invasive means of identifying and monitoring HD disease progression.

## 1. Introduction

Huntington’s disease (HD) is an autosomal-dominant, progressive neurodegenerative disorder that is caused by a CAG-repeat expansion in the Huntington (*HTT*) gene [1]. The most notable symptoms of HD, i.e., chorea, movement dysfunction, cognitive impairment, and behavioral disturbances, result from degeneration in the brain, most strikingly in the striatum and cerebral cortex [2]. However, the encoded Huntingtin protein (Htt) exhibits ubiquitous expression outside of the brain and in many peripheral tissues [3]. Accordingly, it is now clear that the pathology of HD is not exclusively restricted to the brain and several peripheral abnormalities are known to occur in HD patients, including weight loss, altered glucose homeostasis, skeletal muscle wasting and cellular abnormalities [4,5,6]. Comparisons between brain and peripheral sites have suggested that molecular mechanisms through which mutant Htt might lead to cell dysfunction are widely shared between the brain and peripheral tissues [4].

Growing evidence suggests that inflammatory responses, in both the brain and peripheral tissues, play an important role in the pathophysiology associated with HD [7,8]. Specific increases in proinflammatory cytokines, including interleukin 6 (IL-6), interleukin 8, and tumor necrosis factor alpha (TNF-a), have been found in the brains from HD patients and mouse models of HD [9,10,11,12]. Peripheral findings support these brain studies, in that several studies have demonstrated increases in IL-6, C-reactive protein (CRP), and transforming growth factor beta 1, in plasma from human HD patients when compared with controls [10,13,14,15]. Another study found that IL-6 levels in blood were increased in premanifest HD gene carriers prior to the onset of clinical symptoms [11]. IL-6, but not IL-1B and TNF-a, has been detected in human cerebrospinal fluid (CSF), although no significant differences were found between the HD and control subjects [16].

There is growing interest in the ability of salivary inflammatory biomarkers to inform about diagnoses of systemic conditions [17]. Saliva has many advantages over blood as a non-invasive alternative biofluid, including the ease of sample collection in vulnerable patient populations, lack of need for immediate sample processing, and ability to collect samples in diverse settings. In this study, we measured CRP, IL-6, IL-1B, as well as alpha amylase (AA) in both saliva and plasma from premanifest HD, manifest HD, and normal control subjects. Further, we investigated whether these markers were associated with disease data and clinical measures. Biomarkers reflecting these peripheral and/or central derangements of neuroinflammation could be useful to better characterize disease progression and the therapeutic response to specific interventions.

## 2. Results

### 2.1. Participant Characteristics

We collected saliva and plasma samples from 125 individuals, including premanifest patients (“PM”; *n* = 36), manifest patients (“HD”; *n* = 37), and control subjects (“NC”; *n* = 52) (Table 1). The PM patients were significantly younger than control and manifest HD patients (Kruskal–Wallis test, *p* < 0.0001; Table 1). Hence, age-adjustments were implemented in subsequent correlation analyses. Gender distribution was not significantly different between cohorts ((χ^2^ = 0.863, *p* = 0.352) and there was no significant difference in years of education (Table 1). There was a significant difference in weight across cohorts (Kruskal–Wallis test, *p* = 0.024), with HD patients weighed significantly less, a metabolic hallmark of HD. The CAG mutation lengths were not significantly different between the PM and HD groups, nor was parental age of onset (Table 1).

### 2.2. Measures of Inflammatory-Related Markers in Saliva and Plasma

We quantified the levels of IL-6, IL-1B, CRP, and AA in HD, PM, and control subjects using immunoassays. No statistically significant age or sex effects were detected for any of the analytes in control and HD patients, when each cohort was analyzed separately. We compared the levels of all analytes across the three diagnostic groups using a Kruskal–Wallis test with Dunn’s post-test (IL-6 and CRP) or ANOVA with Bonferroni’s post-test (IL-1B and AA). For both IL-6 and IL-1B, the HD patients exhibited significantly higher levels when compared to PM patients, but not compared to NC subjects (IL-6, log pg/mL = 1.42 ± 0.51 vs. 1.00 ± 0.43; *p* < 0.05; IL-1B, log pg/mL= 2.71 ± 0.41 pg/mL vs. 2.53 ± 0.29 *p* < 0.05) (Figure 1). For CRP, PM patients showed significantly higher levels as compared to NC subjects (log pg/mL 3.53 ± 0.61 vs. 3.31 ± 0.34; *p* < 0.05) (Figure 1). Although levels of CRP were higher in HD patients, the difference compared to the NC group was not significantly different (Figure 1).

CRP and IL-6 were also measured in the plasma of HD, PM, and normal controls subjects. Plasma levels of IL-6 were elevated in HD patients as compared to healthy controls (log pg/mL= −0.143 ± 0.278 vs. −0.491 ± 0.432; *p* < 0.01) (Figure 2). The plasma levels of CRP were not significantly different across any subject group (Figure 2).

We next compared associations of all salivary and plasma markers with one another using Spearman correlation analyses. The levels of salivary IL-6 in all subjects were significantly correlated with each of the other salivary markers, IL-1B (rho = 0.527; *p* < 0.0001), CRP (rho = 0.377; *p* = 0.008, and AA (rho = 0.393; *p* = 0.0049), as well as with IL-6 levels measured in plasma (rho = 0.590; *p* < 0.0001) (Table 2; Figure 3). Salivary IL-1B was modestly correlated with AA (rho = 0.266; *p* = 0.027) (Table 2). Similarly, the salivary CRP levels were modestly significantly correlated with plasma levels of IL-6 (rho = 0.254; *p* = 0.047) and plasma CRP (rho = 0.338; *p* = 0.007) (Table 2; Figure 3).

### 2.3. Associations Between Salivary Inflammatory Markers and Clinical Symptoms

We next examined the correlations between inflammatory markers and clinical characteristics in all gene (+) mutation carriers (i.e., PM and HD patients together). The salivary levels of IL-6 were significantly correlated with several clinical features, including Montreal Cognitive Assessment (MoCA), Symbol Digit Modalities (SDM), Total Functional Capacity (TFC), Total Motor Score (TMS), and Total Chorea Score (Chorea) (Table 3), with all but MoCA and SDM remaining significant after age-adjustment (Table 3). The salivary IL-1B levels were also significantly correlated with TFC and TMS, even after correcting for age (Table 3). No significant correlations were observed for salivary CRP or AA with or without age correction. Age-adjusted correlations for plasma CRP were observed for SDM and TFC, but no significant associations were observed for plasma IL-6 (Table 3). No significant associations were observed between any of the analytes and the Mini-Mental State Examination (MMSE) or the Hospital Anxiety and Depression Scale (HADS) (Table 3).

Cognitive and behavioral measurements were also assessed for associations to saliva and plasma markers in the control subjects (Appendix A). In saliva, IL-6 was the only marker significantly correlated with any clinical measures, and these included significant negative correlations with MoCA (age-adjusted rho = −0.500 *p* = 0.021) and SDM (age-adjusted rho = −0.729; *p* < 0.0001) scores (Figure 4). With regards to plasma, neither IL-6 nor CRP levels were significantly correlated with any clinical data in normal control neither subjects (data not shown).

## 3. Discussion

Researchers have increasingly turned to peripheral fluids for biomarker studies on neurodegenerative diseases, including HD. Growing evidence suggests that inflammatory responses occur in both the brain and peripheral tissues in HD. Hence, the identification of inflammation-related peripheral biomarkers might inform regarding HD pathophysiology, as well as possible disease-modifying treatments targeting inflammation. In this study, we measured levels of the inflammatory markers, IL-6, IL-1B, CRP, and AA, in saliva and plasma in HD patients at different stages of illness in addition to control subjects. We detected differences not only among disease groups, but also in association with clinical measures.

We detected increases in IL-6 in both saliva and blood from HD patients compared to either PM or normal subjects. The increase in plasma IL-6 in HD patients is consistent with several previous studies [10,11,13], although we did not detect this increase in PM patients compared to controls, as was shown previously [11]. Elevated levels of IL-6 were also evident in saliva samples from HD patients compared to PM and NC groups, but only the difference between HD and PM reached statistical significance. Perhaps more interesting than diagnostic group differences in IL-6 levels, were the significant correlations observed between salivary IL-6 and several clinical features of HD, including motor and cognitive tests. Significant positive correlations were found between salivary IL-6 and two motor function measures, TMS and chorea, while negative correlations were detected against TFC. Salivary IL-6 was also significantly negatively correlated to SDM, which appraises working memory, in both HD patient groups and control subjects. This finding is consistent with a report showing that plasma IL-6 was associated a cognitive score derived in part from SDM scores, in HD patients [14].

We did not observe similar correlations between plasma IL-6 and clinical data, despite salivary and plasma levels showing a significant correlation in this study. Significant correlations between salivary and plasma IL-6 have been reported previously [18,19], but not all studies have shown significant correlations [20,21]. Interestingly, similar to all of these previous studies, we found that IL-6 levels were much higher in saliva (~20 fold) than in plasma; this could be due to increased activity of the salivary glands in response to certain stimuli [22]. Of note is another study that demonstrated significant correlations between plasma and CSF levels of IL-6 [23], suggesting CNS relevance to blood levels of IL-6.

IL-6 is a pleiotropic cytokine, which triggers the acute phase response by the induction of a systemic reaction consisting of hepatic release of CRP and the release of other pro-inflammatory cytokines, such as IL-1B. Hence, it is perhaps not surprising that levels of salivary IL-6 were correlated with these two analytes in saliva and that both CRP and IL-1B showed elevated levels when compared across disease groups. These findings are consistent with a previous study showing robust correlations between salivary cytokines in individual saliva samples [24].

In addition to its role in the acute phase response, IL-6 also exerts endocrine and metabolic effects on various organs including liver, fat and skeletal muscle. In particular, IL-6 is released from normal skeletal muscle in response to exercise [25], mediating anti-inflammatory responses and metabolic adaptations, actions contradictory to the typical view that IL-6 is a proinflammatory cytokine that is inducing and/or propagating disease pathology [26]. Chorea, the hallmark symptom of HD, consists of excessive, irregularly-timed spontaneous movements. Studies have shown that HD skeletal muscle is hyperexcitable, which can cause involuntary and prolonged contractions that may contribute to chorea [27] and could produce elevated levels of IL-6. However, it is also known that chronically elevated IL-6 production can promote skeletal muscle wasting [28,29] and muscular atrophy [30]. In addition to the general skeletal muscle dysfunction occurring in HD [5,31,32,33], skeletal muscle wasting and/or atrophy is likely to represent a significant component of HD pathogenesis [31]. In either case, IL-6 is likely to play an important role in local regulation of the inflammatory process in skeletal muscle in HD.

The increases in the levels of IL-6 and IL-1B in saliva from HD patients detected in this study mimic the elevated levels of cytokines reported in the human HD brain [9,10,11]. However, it is unlikely that salivary levels results from passage of cytokines from the CNS into blood, then subsequently into saliva, given that cytokines are rapidly broken down and/or would be substantially diluted by the time they reached saliva. Increased levels of IL-6 could result from increased synthesis by leukocytes or buccal epithelial cells, which are both present in saliva. Alternatively, elevated IL-6 could be attributed to increased secretion from the acinar and duct cells of the salivary glands. The secretion of salivary fluid and proteins is controlled by autonomic nerves [34] and previous studies have suggested that increased sympathetic activity could be attributed to prolonged increases in salivary IL-6 levels following acute psychosocial stress [22]. Sympathetic hyperfunction has been shown to be present in HD patients at different stages of illness [35,36], suggesting further relevance of this system in IL-6 secretion.

CRP, a marker for systemic inflammation, was increased in PM patients compared to normal controls, which is similar to previous studies showing elevated CRP levels in blood samples from pre-symptomatic HD patients vs. controls [15]. However, we did not observe a significant increase in CRP in blood samples, although levels were higher in PM patients compared to controls, the difference was not statistically significant. This finding was somewhat surprising, given that salivary and plasma levels of CRP were significantly correlated with one another, similar to what has been observed in previous studies [37].

AA is the most abundant protein in saliva, and previous studies have shown that AA can serve as a biomarker for local, oral inflammation [38,39], however, we did not detect any differences in AA levels across disease groups in this study, suggesting that changes observed in HD were not due to differences in local oral health. However, we cannot rule out the possibility that the increases in salivary IL-6 and IL-1B might be associated with a local oral immune response in HD patients.

Biomarker research in saliva has grown over the past few years [40], largely due to its non-invasive nature and ease of collection. In particular, there is a growing interest in studying cytokines in oral fluids, with IL-6 being one of the most widely investigated cytokines. One important consideration when assessing salivary cytokines is the type of oral fluid used. This study used whole saliva, which is a complex mixture of secretions from several salivary glands, while other studies have used oral mucosal transudate (OMT) (also called gingival crevicular fluid), which is the fluid derived from the transport of serum components through the oral mucosa. These fluids differ in their levels of cytokines, which is largely dependent on the presence of serum constituents in OMT [41]. Other factors contributing to differing analyte levels in saliva include smoking, stress and medications [42], which should be considered in future studies.

In summary, this study is the first to measure salivary inflammation markers in the context of HD and we highlight salivary levels of IL-6 as being significantly associated with prominent disease symptoms in our HD mutation carriers, as well as being correlated to cognitive measures in normal participants. These results suggest that inflammatory changes detected in peripheral saliva may be biologically relevant and mirror the neurodegenerative process occurring in the CNS. Although IL-6 elevation would not likely be specific to HD compared to other neurodegenerative diseases, the advent of an effective, dependable salivary biomarker would meet the urgent need for a less invasive means of identifying and monitoring HD disease progression.

## 4. Materials and Methods

### 4.1. Human Subjects

This study was approved by the UCSD Institutional Review Board (#170038, approved on 14 December 2019), in accordance with the requirements of the Code-of-Federal-Regulations on the Protection of Human Subjects. Patients were recruited from the University of California, San Diego HDSA Center of Excellence and included a diagnosis of HD with family history. Normal controls had no reported history of neurological conditions, psychiatric disorders and no use of psychoactive substances. All participants gave informed consent prior to sample collection. Demographic and disease data was collected at the time of saliva collection, including sex, age, CAG-repeat length, education, and family history. Disease burden scores (DBS) were calculated from this information, which is provided in Table 1.

### 4.2. Clinical Assessments

All study participants underwent clinical assessments, including cognitive testing, behavioral and functional measures, and motor ratings. The cognitive battery included the Mini-Mental State Examination (MMSE), Montreal Cognitive Assessment (MoCA) and Symbol Digit Modalities test (SDM). Psychiatric changes were assessed using Hospital Anxiety and Depression Survey (HADS). Functional proficiency was evaluated using the UHDRS Total Functional Capacity (TFC). Motor dysfunction was assessed using the UHDRS Total Motor Score (TMS). The sum of all maximal chorea sub-scores were also noted and referred to as “Total Chorea”. These data are summarized in Appendix A.

### 4.3. Plasma Collection

Blood from consenting HD patients and controls was drawn by venipuncture into 2 mL lavender/EDTA tubes. EDTA/whole blood was mixed well by inversion and spun at 900× *g* for 15 min. The top plasma layer was transferred into 1-mL aliquots and snap frozen and stored at −80 °C. All subjects who provided a plasma sample, also provided a saliva sample.

### 4.4. Saliva Collection

All donors were asked to refrain from smoking, eating, drinking, or oral hygiene procedures for at least 1 h prior to samples collection. Saliva samples were collected between 10 a.m.–4 p.m. using the passive drool method according to previously established protocols [43]. Roughly two milliliters of unstimulated whole saliva was obtained. Besides the patients who provided a plasma sample, an additional group of patients provided a saliva-only sample. Samples were immediately frozen at −20 °C at the time of collection, then stored at −80 °C. At the time of use, saliva samples were thawed and centrifuged (10,000× *g*; 10 min; 4 °C) to remove insoluble material and cellular debris. Supernatants were collected and used for all assays.

### 4.5. ELISA Assays

Salivary levels of CRP, IL-6, IL-1B, and alpha-amylase were measured run by Salimetrics, LLC (Carlsbad, CA, USA). The amount of sample used per well and assay ranges were as follows: CRP: 50 µL; 19.44−1600 pg/mL; IL-6: 60 µL; 0.35−500 pg/mL; IL-1β: 20 µL; 5.55−1000 pg/mL; AA: 10 µL; 0.4−1600 U/mL. Plasma levels of CRP, IL-6, IL-1B were measured using commercially available ELISA (CRP, R&D Systems; IL-6 and IL-1B, Abcam), according to the manufacturers’ instructions. IL-6 and IL-1B assays used 50 µL plasma per well, while plasma was diluted 100× for the CRP assay. Assays ranges were 0.78−50 ng/mL for CRP; 1.56−100 pg/mL for IL-6; and 1.95−125 pg/mL for IL-1B. The measurements of IL-1B in plasma were below the detection limit of our assay, hence not reported. AA levels are not found at appreciable amounts in circulation, so were not measured in plasma samples. All of the assays were performed in duplicate by operators blinded to the clinical state of the participant. The same salivary analytes were not measured in all of the subjects, due to different recruitment time points.

### 4.6. Statistics

The distribution of the data values for each analyte was tested for normality using the Kolmogorov–Smirnov normality test. Most of the analytes data did not show normal distribution; hence, logarithm transformation was applied to improve normality. Outlier analyses of the log-transformed data values were carried out using the Iglewicz and Hoaglin’s test for multiple outliers (two-sided-test; z = 3.5) for diagnostic comparisons, resulting in the removal of one datapoint for the plasma CRP, two datapoints for the plasma IL-6 and three datapoints for the saliva IL-6, and these values were removed for subsequent analyses. Demographic and clinical characteristics were compared between study groups using chi-square tests, analysis of variance with post hoc Bonferroni tests or Kruskal–Wallis tests followed by Mann–Whitney U tests, where appropriate. For comparing levels of each saliva or plasma analyte against clinical variables, partial correlations analysis (both parametric and nonparametric) were carried out adjusting for age. MMSE and MoCA were normally distributed, while the remaining measures, TMS, chorea scores, PBA, and TFC, were not normally distributed. The SDM scores were normally distributed in normal controls, but not in HD patients. Statistical analyses were performed using SPSS Statistics (partial correlation analyses) and Prism (all other analyses).

## Figures and Tables

**Figure 1 ijms-21-06363-f001:**
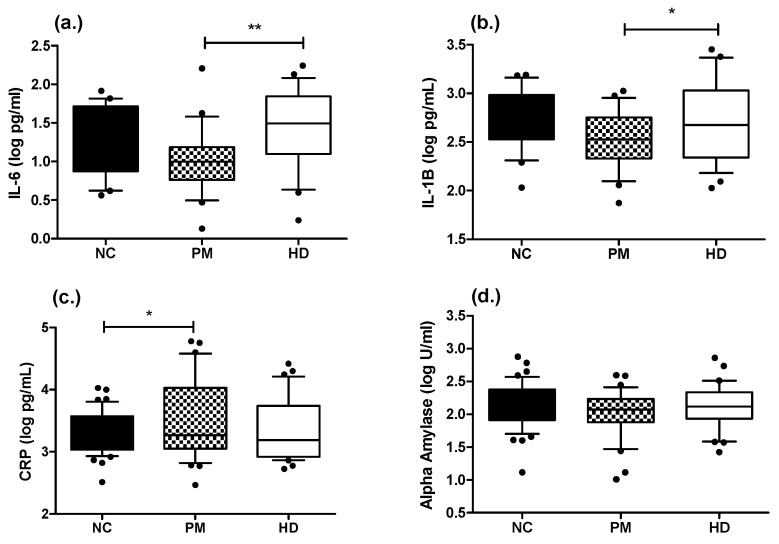
Comparisons of salivary levels of IL-6, IL-1B, CRP, and AA in Huntington’s patients and control subjects. Significant differences in levels of IL-6 (panel **a**), IL-1B (panel **b**), CRP (panel **c**) and AA (panel **d**) across diagnostic groups were determined by ANOVA with Bonferroni’s post-test. Post-test p-values: ** *p* < 0.01 and * *p* < 0.05. Box plots represent median and interquartile range, with whiskers at 10−90 percentiles. NC, normal control (*n* = 22 for IL-6 and IL-1B, *n* = 44 for CRP and *n* = 47 for AA); PM, premanifest (*n* = 24 for IL-6 and IL-1B, *n* = 35 for CRP and *n* = 32 for AA); HD, manifest Huntington’s disease (*n* = 22 for IL-6, *n* = 24 for IL-1B, *n* = 33 for CRP and *n* = 36 for AA).

**Figure 2 ijms-21-06363-f002:**
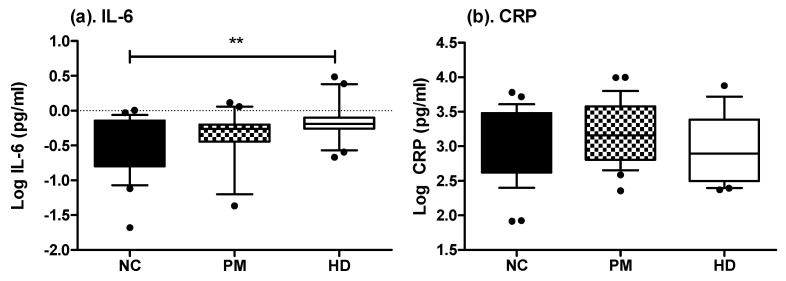
Comparisons of plasma levels of IL-6 and CRP in HD patients and control subjects. Significant differences in plasma levels of IL-6 (**a**) and CRP (**b**) across diagnostic groups were determined by ANOVA, with Bonferroni’s post-test: ** *p* < 0.01. Box plots represent median and interquartile range, with whiskers at 10−90 percentiles. NC, normal control (*n* = 27 for IL-6 and *n* = 29 for CRP); PM, premanifest (*n* = 29 for IL-6 and 26 for CRP); HD, manifest Huntington’s disease (*n* = 21 for IL-6 and *n* = 22 for CRP).

**Figure 3 ijms-21-06363-f003:**
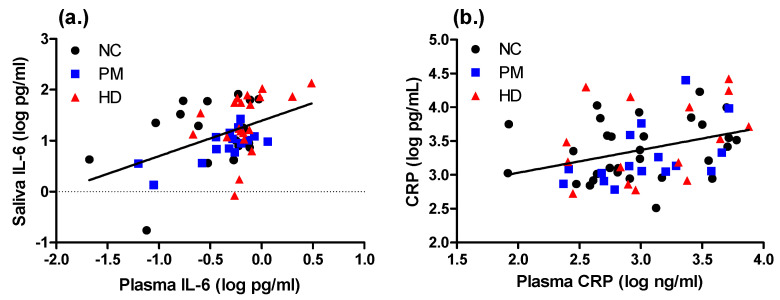
Correlations between plasma and saliva levels of IL-6 and CRP. Levels of IL-6 and CRP are significantly correlated in matched plasma and saliva samples. Panel (**a**) shows correlations for IL-6: rho = 0.590; *p* < 0.0001, for log-transformed values. NC, normal control (*n* = 18; black circles); PM, premanifest (*n* = 16; blue triangles); HD, manifest Huntington’s disease (*n* = 19; red circles). Panel (**b**) shows correlations for CRP: rho = 0.338; *p* = 0.007, for log-transformed values. NC, normal control (*n* = 30; black circles); PM, premanifest (*n* = 16; blue triangles); HD, manifest Huntington’s disease (*n* = 16; red circles).

**Figure 4 ijms-21-06363-f004:**
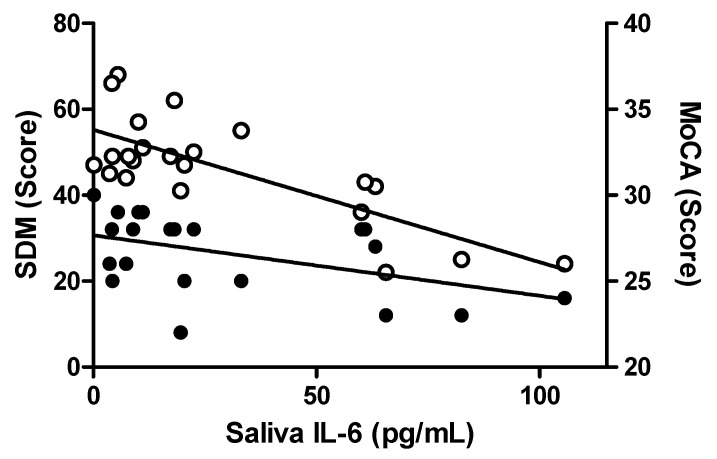
Correlations between salivary IL-6 levels and cognitive measures in normal control subjects. Levels of salivary IL-6 are significantly negatively correlated to SDM (Symbol Digit Modality Score; left y-axis; open circles; Age-adjusted Pearson rho = −0.729; *p* < 0.0001) and to MoCA (Montreal Cognitive Assessment; right y-axis; closed circles; Age-adjusted Pearson rho = −0.500; *p* = 0.021). Data for SDM and MoCA were only available for a subset of control subjects (*n* = 22).

**Table 1 ijms-21-06363-t001:** Demographic and disease data for subjects used in this study, mean (range).

	NC	PM	HD	*p*-Value
**n**	52	36	37	
**Gender, F:M**	27:25	19:17	25:12	0.352
**Age, yrs**	54.42 (23−78)	43.44 (19−71) **	56.95 (30−76)	<0.0001
**Weight, lbs**	171.94 (106−306)	172.90 (99−290)	149.83 (90−263) *^	0.024
**Education, yrs**	15.40 (12−22)	16.24 (12−22)	14.22 (5−24)	0.438
**CAG repeat**	NA	42.33 (38−51)	42.93 (38−49)	0.283
**DBS**	NA	248.39 (112−368)	388.14 (273−578) ^^	<0.0001
**AOO, yrs**	NA	NA	48.83 (22−68)	NA
**PAO, yrs**	NA	48.91 (24−75)	50.16 (27−70)	0.222

*p*-values (*p*) reflect the following statistical tests: Gender, Chi-square test; Age, Weight and Education, Kruskal–Wallis ANOVA; CAG repeat, DBS and PAO, Mann–Whitney U test. For ANOVA, the individual comparisons were determined by Tukey post-hoc analysis as follows: * *p* < 0.05 between NC and HD; ** *p* < 0.001 between PM and both NC and HD; ^ *p* < 0.05 between PM and HD; ^^ *p* < 0.01 between PM and HD. NC, Normal control; PM, Pre-manifest Huntington’s disease; HD, Manifest Huntington’s disease; DBS, Disease Burden Scale; AOO, Age of onset; PAO, Parental age of onset; NA, Not applicable; ND, Not determined.

**Table 2 ijms-21-06363-t002:** Correlations across salivary and plasma markers.

		Saliva IL-6 (pg/mL)	Saliva IL-1B (pg/mL)	Saliva CRP (pg/mL)	Saliva AA (U/mL)	Plasma IL-6 (pg/mL)	Plasma CRP (pg/mL)
Saliva IL-6 (pg/mL)	Spearman’s rho	—					
	p-value	—					
Saliva IL-1B (pg/mL)	Spearman’s rho	0.527 ***	—				
	p-value	**<0.0001**	—				
Saliva CRP (pg/mL)	Spearman’s rho	0.377**	0.06	—			
	p-value	**0.008**	0.633	—			
Saliva AA (U/mL)	Spearman’s rho	0.393 **	0.266*	0.099	—		
	p-value	**0.006**	**0.027**	0.378	—		
Plasma IL-6 (pg/mL)	Spearman’s rho	0.381 **	0.109	0.254 *	0.029	—	
	p-value	**0.0049**	0.405	**0.047**	0.815	—	
Plasma CRP (ng/mL)	Spearman’s rho	0.165	0.115	0.345 **	−0.134	0.135	—
	p-value	0.181	0.358	**0.006**	0.266	0.274	—

Spearman correlations analyses were carried out for all comparisons. Bold values indicate statistically significant correlations as indicated. * *p* < 0.05; ** *p* < 0.01; *** *p* < 0.0001.

**Table 3 ijms-21-06363-t003:** Correlations between inflammatory markers and clinical symptoms in Huntington’s patients.

Unadjusted Correlations and *p*-Values												
	Saliva IL6 (pg/mL)		Saliva IL1B (pg/mL)		Saliva CRP (pg/mL)		Saliva AA (U/mL)		Plasma IL6 (pg/mL)		Plasma CRP (ng/mL)	
	rho	*p*-Value	rho	*p*-Value	rho	*p*-Value	rho	*p*-value	rho	*p*-value	rho	*p*-Value
MMSE	−0.219	0.154	−0.194	0.191	−0.177	0.243	0.106	0.479	−0.215	0.183	0.041	0.786
MoCA	**−0.317**	**0.036**	**−0.317**	**0.03**	−0.02	0.897	−0.004	0.98	−0.117	0.472	0.027	0.86
SDM	**−0.415**	**0.006**	−0.226	0.135	0.007	0.965	0.048	0.758	−0.177	0.337	**0.341**	**0.027**
TFC	**−0.477**	**0.001**	**−0.348**	**0.016**	0.069	0.652	−0.19	0.201	−0.289	0.07	0.224	0.134
HADS	−0.047	0.767	0.062	0.686	0.07	0.654	−0.136	0.373	0.027	0.87	−0.02	0.902
TMS	**0.51**	**< 0.000**	**0.38**	**0.008**	−0.081	0.598	0.143	0.339	0.258	0.108	−0.166	0.27
Chorea	**0.549**	**< 0.000**	**0.333**	**0.022**	−0.038	0.803	0.221	0.135	0.152	0.35	−0.203	0.181
**Age-Adjusted Correlations and *p*-Values**												
	**Saliva IL6 (pg/mL)**		**Saliva IL1B (pg/mL)**		**Saliva CRP (pg/mL)**		**Saliva AA (U/mL)**		**Plasma IL6 (pg/mL)**		**Plasma CRP (ng/mL)**	
	**rho**	***p*-Value**	**rho**	***p*-Value**	**rho**	***p*-Value**	**rho**	***p*-Value**	**rho**	***p*-Value**	**rho**	***p*-Value**
MMSE	−0.124	0.428	−0.155	0.305	−0.174	0.258	0.133	0.377	−0.178	0.278	0.054	0.724
MoCA	−0.242	0.118	−0.269	0.071	−0.011	0.943	0.018	0.906	−0.078	0.639	0.039	0.805
SDM	−0.232	0.144	−0.153	0.322	0.036	0.821	0.123	0.433	−0.095	0.576	**0.457**	**0.003**
TFC	**−0.323**	**0.034**	**−0.306**	**0.038**	0.111	0.474	−0.174	0.247	−0.237	0.146	**0.303**	**0.043**
HADS	−0.173	0.281	0.017	0.911	0.063	0.692	−0.163	0.289	0.075	0.66	−0.031	0.847
TMS	**0.35**	**0.011**	**0.356**	**0.015**	−0.141	0.363	0.117	0.439	0.197	0.23	−0.258	0.088
Chorea	**0.411**	**0.006**	0.29	0.051	−0.079	0.611	0.218	0.146	0.058	0.725	−0.293	0.053

Parametric and nonparametric Partial correlation analyses were carried out, unadjusted (top) and age-adjusted (bottom). Bold values indicate statistically significant correlations. MMSE, Mini-Mental State Exam; MoCA, Montreal Cognitive Assessment; SDM, Symbol Digit Modalities test; TFC, Total Functional Capacity; HADS, Hospital Anxiety and Depression Score; TMS, Total Motor Score; Chorea reflects Total Chorea Score. Significant associations are shown in bold.

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
