# Peer review of "Levels of Interleukin-6 in Saliva, but Not Plasma, Correlate with Clinical Metrics in Huntington’s Disease Patients and Healthy Control Subjects"

_ijms, 2020, doi:10.3390/ijms21176363_

Round 1

Reviewer 1 Report

Ref: ijms-919398

Title: Levels of interleukin-6 in saliva, but not plasma, correlate with clinical metrics in Huntington’s disease patients and healthy control subjects (Journal: IJMS)

Recommendation: Minor Revision

Comments:

  1. What would be the specificity of IL-6 as a HD biomarker compared to other neurodegenerative diseases? Could this parameter possibly also be a target of therapy?
  2. The “n” number should be added to each Figure.
  3. Does the statistical significance in Figure 1 (IL-6 in saliva) result from the so-called outliners, e.g. two HD patients who have IL-6 ~150 pg / ml or one PM patient with IL-6 in saliva ~ 150 pg/ml?
  4. Does the statistical significance in Figure 2 (IL-6 in plasma) result from three HD patients who have IL-6 over 1 pg/ml?
  5. Lines 117-118 – where in Figure 2 are the IL-6 results 8.55±7.41 pg/ml vs. 4.58±3.06 pg/ml.

Author Response

  What would be the specificity of IL-6 as a HD biomarker compared to other neurodegenerative diseases? Could this parameter possibly also be a target of therapy?

We thank the reviewer for raising these excellent points.  We have added a few sentences to the Discussion (lines 196-197 and lines 282-283) to address the specificity of IL-6 and its potential to monitor therapeutic efficacy.

  The “n” number should be added to each Figure.

This has been added.

  Does the statistical significance in Figure 1 (IL-6 in saliva) result from the so-called outliners, e.g. two HD patients who have IL-6 ~150 pg / ml or one PM patient with IL-6 in saliva ~ 150 pg/ml? 

Outlier analyses were carried out using the Iglewicz and Hoaglin's test for multiple outliers (two-sided-test; z=3.5), which resulted in the removal of three datapoints for the saliva IL-6 dataset.  After removing these outliers, three of the IL-6 values still appear higher than others, as the reviewer mentions, although these are not outliers.  However, if we did remove the high values in question (“two HD patients who have IL-6 ~150 pg / ml or one PM patient with IL-6 in saliva ~ 150 pg/ml”), the difference between PM and HD is still statistically significant (One-way ANOVA p=0.0083; Bonferroni post-test, PM vs HD, p<0.01).  Due to the variability in our data, we used log-transformed data in our outlier analyses, which greatly reduced the number of outliers that we observed.  Accordingly, we now show log-transformed data in our graphs, which improves the appearance.

 Does the statistical significance in Figure 2 (IL-6 in plasma) result from three HD patients who have IL-6 over 1 pg/ml?

Similar to the saliva IL-6 data, outlier analyses of the log-transformed data were carried out using the Iglewicz and Hoaglin's test for multiple outliers (two-sided-test; z=3.5); this resulted in the removal of two datapoints for the plasma IL-6, which were already removed from analysis.  However, if we did remove all values over 1 pg/ml (although these were not outliers), the difference between HD and NC would still be statistically significant (One-way ANOVA p=0.05; Bonferroni post-test, PM vs HD, p<0.05).  As above, we now show the log-transformed data in our graphs.

  Lines 117-118 – where in Figure 2 are the IL-6 results 8.55±7.41 pg/ml vs. 4.58±3.06 pg/ml.

We apologize for the error in these values and they have now been corrected.

Reviewer 2 Report

This is a very well written and interesting study that advances the field.

The design and statistics are good (n is a bit low but satisfactory). Results are clearly presented and conclusions are consistent with the results. The authors do a fine job of discussing the findings in the context of the current literature.

There are no major concerns.

Author Response

We thank the reviewer for his positive assessment of our manuscript and very please that he/she has no major comments.  We have carried out a Spell check to correct minor typos, as requested.